# Integrated Analysis of the Altered lncRNAs and mRNAs Expression in 293T Cells after Ionizing Radiation Exposure

**DOI:** 10.3390/ijms20122968

**Published:** 2019-06-18

**Authors:** Mengmeng Yang, Yuxiao Sun, Changyan Xiao, Kaihua Ji, Manman Zhang, Ningning He, Jinhan Wang, Qin Wang, Zhijuan Sun, Yan Wang, Liqing Du, Yang Liu, Chang Xu, Qiang Liu

**Affiliations:** Tianjin Key Laboratory of Radiation Medicine and Molecular Nuclear Medicine, Institute of Radiation Medicine, Chinese Academy of Medical Sciences and Peking Union Medical College, Tianjin 300192, China; yangmengmeng62@163.com (M.Y.); syx17525@163.com (Y.S.); xiaochangyan39@163.com (C.X.); jikaihua@irm-cams.ac.cn (K.J.); zhangmanman@irm-cams.ac.cn (M.Z.); hening1123@126.com (N.H.); wangjinhan@irm-cams.ac.cn (J.W.); wangqin@irm-cams.ac.cn (Q.W.); sunzj@irm-cams.ac.cn (Z.S.); wangyan@irm-cams.ac.cn (Y.W.); duliqing@irm-cams.ac.cn (L.D.); liuyang@irm-cams.ac.cn (Y.L.)

**Keywords:** ionizing radiation, DNA damage, lncRNA, mRNA

## Abstract

Tissue and cell damage caused by ionizing radiation is often highly genotoxic. The swift repair of DNA damage is crucial for the maintenance of genomic stability and normal cell fitness. Long noncoding RNAs (lncRNAs) have been reported to play an important role in many physiological and pathological processes in cells. However, the exact function of lncRNAs in radiation-induced DNA damage has yet to be elucidated. Therefore, this study aimed to analyze the potential role of lncRNAs in radiation-induced DNA damage. We examined the expression profiles of lncRNAs and mRNAs in 293T cells with or without 8 Gy irradiation using high-throughput RNA sequencing. We then performed comprehensive transcriptomic and bioinformatic analyses of these sequencing results. A total of 18,990 lncRNAs and 16,080 mRNAs were detected in all samples. At 24 h post irradiation, 49 lncRNAs and 323 mRNAs were differentially expressed between the irradiation group and the control group. qRT-PCR was used to verify the altered expression of six lncRNAs. Gene Ontology (GO) and Kyoto Encyclopedia of Genes and Genomes (KEGG) analyses indicated that the predicted genes were mainly involved in the histone mRNA metabolic process and Wnt signaling pathways. This study may provide novel insights for the study of lncRNAs in radiation-induced DNA damage.

## 1. Introduction

Genome integrity is crucial for all living cells and organisms. However, ionizing radiation (IR) gives rise to widespread DNA damage either directly or through IR-induced reactive oxygen species (ROS) in our cells, endangering genome integrity. If damaged DNA in cells is left unrepaired or is inaccurately repaired, it may result in loss of genetic information, leading to serious clinical consequences [1], including neurodegenerative changes, infertility, immune deficiency, and cancer [2,3]. It has become clear that molecular cell signaling plays a crucial role in determining the cellular response to radiation. To mitigate the threat of DNA damage, cells have evolved elaborate protective mechanisms to sense DNA damage, present damage signals, and eliminate damage by regulating cellular responses [4,5]. This process is referred to as the DNA damage response (DDR) [4,5,6]. The cellular response to ionizing radiation involves regulatory changes in many processes, including transcription, mRNA processing, and translation. 293T cells are derived from human embryonic kidney 293 cells, and constitutively express SV40 large T antigen, which allows for the replication of plasmids containing the SV40 origin of replication [7,8]. Due to the high transfection efficiency, 293T cells are one of the most commonly used cell lines in the studies of DDR [9,10,11,12,13].

Long noncoding RNAs (lncRNAs) are a group of noncoding RNA transcripts with a length greater than 200 nucleotides [14]. Approximately 75% of the human genome is transcribed, and the majority of the transcripts are annotated as noncoding RNAs (ncRNAs) [15]. These ncRNAs are subdivided into two main categories depending on their length. Small ncRNAs (sncRNAs) are shorter than 200 nucleotides, and long ncRNAs (lncRNAs) are longer than 200 nucleotides. lncRNAs have long been considered to represent transcriptional noise because they do not encode proteins [16]. In recent years, however, accumulating evidence has supported the notion that lncRNAs play important regulatory roles in various biological functions, including transcriptional activation, transcriptional interference, histone modification, chromatin remodeling, cell cycle regulation, epigenetics, and RNA splicing [17,18,19]. Moreover, lncRNAs have been reported to exert some disease-related effects; for example, lncRNAs exhibit a high degree of disorder in cancer and serve as potential biomarkers for diagnosis, treatment, and prognosis of cancer [16,20]. In addition, lncRNAs are associated with infectious diseases and ophthalmological diseases [21,22].

Recent studies have shown that after exposure to ionizing radiation, some lncRNAs are involved in the DDR by repairing damaged DNA. For instance, lncRNA LIRR1 is reported to regulate the DDR by increasing radiosensitivity, arresting the cell cycle at G1 phase and increasing γ-H2AX foci [23]. lncRNA LINP1 can serve as a scaffold that links Ku80 and the catalytic subunit of DNA-dependent protein kinase (DNA-PKcs), coordinating the non-homologous end joining (NHEJ) pathway. Moreover, it can regulate p53 and epidermal growth factor receptor (EGFR) signaling, which increases the sensitivity of the tumor cell response to radiotherapy in breast cancer [24]. lncRNA HOTAIR can act as a scaffold for two distinct histone modification complexes. lncRNA lnc-RI (long noncoding RNA-radiation induced) acts as a competitive endogenous RNA (ceRNA) to stabilize RAD51 mRNA by competitive binding with miR-193a-3p and the release of its inhibition of RAD51 expression [25]. lncRNA NORAD shows a high degree of evolutionary conservation in mammals and is an abundant lncRNA that is indirectly regulated by p53 [26]. Upon DNA damage, NORAD can separate PUMILIO proteins and make them unavailable to repress their target genes, which are critical for mitosis, DNA repair and DNA replication. In contrast, in the absence of NORAD, there is increased chromosomal instability, leading to aneuploidy [26,27]. lncRNA DDSR1 is a DNA damage-inducible lncRNA that positively regulates homologous recombination (HR) repair by promoting the correct accumulation of BRCA1 and RAP80 at double-strand break (DSB) sites and interacting with BRCA1 and hnRNPUL1 [28].

Past studies have shown that a few lncRNAs are involved in DNA damage induced by ionizing radiation. However, the function and regulatory network of lncRNAs are not fully understood. There is very limited information demonstrating the role of lncRNAs in the radiation-induced DDR. Therefore, we tried to identify lncRNAs involved in the radiation-induced DDR to provide accurate therapeutic targets for cancer and biological markers. In this study, high-throughput sequencing was performed to identify the expression profiles of lncRNAs and mRNAs in 293T cells with or without 8 Gy irradiation. Subsequently, we identified differentially expressed mRNAs and lncRNAs. In addition, we performed systematic bioinformatic analyses of Gene Ontology (GO), Kyoto Encyclopedia of Genes and Genomes (KEGG) pathways, and lncRNA–miRNA–mRNA networks to define the potential roles of lncRNAs in the radiation-induced DDR.

## 2. Results

### 2.1. Expression and Classification of lncRNAs in 293T Cells after Exposure to Ionizing Radiation

High-throughput sequencing was used to examine the expression of lncRNAs in the nontreated and 8 Gy-irradiated 293T cells. A total of 18,990 lncRNAs were obtained from all samples (Appendix A). There were collected from the authoritative database Ensembl, TCONS, TUCP, UCSC knownGene, RefSeq, and some data reported in the literature (Figure 1a). According to the relative chromosomal position of the coding gene, lncRNAs can be classified into six broad categories: exon sense-overlapping, intron sense-overlapping, intronic antisense, natural antisense, bidirectional, and intergenic [29]. The 18,990 lncRNAs included 6865 intergenic lncRNAs (36%), 3731 exon sense-overlapping (20%), 2997 natural antisense (16%), 2359 intronic antisense (12%), 1551 intron sense-overlapping (8%), 1487 bidirectional lncRNAs (8%) (Figure 1b).

We further systematically analyzed the expression characteristics of these lncRNAs regarding their distributions in terms of length and on the chromosomes. All lncRNAs ranged in length from 80 to more than 6000 nt, and most of the lncRNAs were 200–3000 nt long (Figure 1c). Chromosome distribution analysis revealed that the 18,990 lncRNAs were located on all chromosomes. Chromosome 1 contained a relatively large number of lncRNAs. Chromosome X exhibited 521 lncRNAs, while chromosome Y contained 15 lncRNAs, and the mitochondrial chromosomes harbored 18 lncRNAs (Figure 1d).

### 2.2. Profiles of Differentially Expressed lncRNAs and mRNAs

After high-throughput sequencing, a total of 18,990 lncRNAs and 16,080 mRNAs were obtained in the control and 8 Gy irradiation groups (Appendix A). The expression profiles of the lncRNAs showed that 2220 lncRNAs only existed in the control group and 1961 lncRNAs only existed in the 8 Gy irradiation group, while 14,809 lncRNAs were common to both groups (Figure 2a). We obtained 25 significantly upregulated lncRNAs and 24 significantly downregulated lncRNAs at 24 h post irradiation compared with the controls *(p* < 0.05, fold change ≥2 ) (Figure 2b and Appendix A).

Regarding the expression profiles of mRNAs, 439 mRNAs were only detected in the control group, and 571 mRNAs existed only in the 8 Gy irradiation group, while 15,070 mRNAs were present in both groups (Figure 2d). Similarly, among all of these mRNAs, compared with the control group, the 8 Gy irradiation group exhibited 227 mRNAs that were significantly upregulated and 96 mRNAs that were significantly downregulated (*p* < 0.05, fold change ≥ 2) (Figure 2e and Appendix A). The scatter plots were used for analyzing the differential expression of lncRNAs and mRNAs between the two groups (Figure 2c,f).

### 2.3. Differentially Expressed lncRNA Validation by Quantitative PCR

We next performed qPCR to validate the high-throughput sequencing results for the differentially expressed lncRNAs. Six differentially expressed lncRNAs were subjected to experimental verification based on random selection. The qRT-PCR results indicated that ENSG00000250519, ENSG00000231595, ENSG00000254338, and BC053669 were upregulated, and ENSG00000223749 and ENSG00000273004 were downregulated after 24 h of 8 Gy ionizing radiation. These results were consistent with our sequencing data (Figure 3).

### 2.4. Functional Annotation of Differentially Expressed lncRNAs and mRNAs

To better understand the biological functions of lncRNAs and mRNAs in 293T cells after radiation, we performed Gene Ontology (GO) analysis on the lncRNAs and mRNAs with significantly altered expression level. The GO terms cover three domains: biological process, cellular component, and molecular function. We found that the most enriched GO terms in these three categories were pre-mRNA binding (ontology:molecular function, GO:0036002), basic amino acid transport (ontology:biological process, GO:0015802), histone mRNA metabolic process (ontology:biological process, GO:0008334), innate immune response in mucosa (ontology:biological process, GO:0002227), and nucleoplasm (ontology:cellular component, GO:0005654) (Figure 4 and Appendix A).

For mRNAs, the GO terms in these three categories mainly included protein heterodimerization activity (ontology:molecular function, GO:0046982), nucleosome organization (ontology:biological process, GO:0034728), DNA bending complex (ontology:cellular component, GO:1990104), and nucleosome (ontology:cellular component, GO:0000786) (Figure 5 and Appendix A).

### 2.5. KEGG Enrichment Analysis

KEGG pathway analysis indicated that after radiation, the mRNA and coding genes adjacent to the identified lncRNAs were mainly associated with the proximal tubule bicarbonate reclamation (hsa04964), glutamatergic synapse (hsa04724), Wnt signaling pathway (hsa04310), Rap1 signaling pathway (hsa04015), transcriptional dysregulation in cancer pathways (hsa05202), and viral carcinogenesis (hsa05203) categories. The significantly enriched pathways are presented in Figure 6 and Appendix A.

### 2.6. lncRNA–miRNA–mRNA Network Analysis

To further understand the possible functions of lncRNAs in the radiation-induced DDR, we constructed the lncRNA–miRNA–mRNA network. Eleven upregulated lncRNAs, four downregulated lncRNAs and 10 differentially expressed mRNAs were selected to build the lncRNA–miRNA–mRNA network. There were 438 nodes and 220 edges between the 10 mRNAs, 175 miRNAs, and 15 lncRNAs. One lncRNA can associate with multiple miRNAs, and one miRNA can inhibit multiple mRNAs. lncRNA ENST00000422178 associated with three different miRNAs, and the target mRNAs included poly(ADP-ribose) polymerase family member 10 (PARP10), LIM zinc finger domain containing 2 (LIMS2), EGF, Disrupted-in-Schizophrenia 1 (DISC1), and H1 histone family member 0 (H1F0). PARP10 has been studied extensively in DNA damage repair [30]. H1F0 is also involved in DNA damage [31] (Figure 7 and Appendix A).

## 3. Discussion

With the rapid development of genome-wide transcriptome analysis, an increasing number of studies have been focusing on the investigation of lncRNAs that take part in the DNA damage response [32]. lncRNAs have been shown to play an important role in DNA damage repair, and lncRNA expression disorders often lead to pathological conditions [33]. However, the mechanism of their contribution to DNA repair pathways remains largely unknown. Therefore, we conducted studies on differentially expressed lncRNAs in the DDR induced by ionizing radiation through high-throughput RNA sequencing.

In this study, we focused on lncRNA and mRNA expression profiles in irradiation-treated 293T cells. By high-throughput RNA sequencing, 18,990 lncRNAs and 16,080 mRNAs were obtained in both nontreated and irradiated groups. Among these lncRNAs, 1961 lncRNAs were detected only in the 8 Gy irradiation group. Furthermore, in this expression profile, 25 lncRNAs were significantly upregulated, and 24 lncRNAs were significantly downregulated in the irradiated group compared with the control group. Additionally, we identified 571 mRNAs that only existed in the 8 Gy irradiation group, and 15,070 mRNAs were detected in both groups. Compared with the control group, 227 mRNAs were significantly upregulated and 96 mRNAs were significantly downregulated in the irradiated group.

In past work, evidence has been found that lncRNAs play a vital role in the regulation of the DDR and are implicated in maintaining genomic stability, both of which are important for cell survival and normal functioning to prevent tumorigenesis [32]. However, studies on lncRNA function are still rare. Therefore, we performed functional analysis of differentially expressed mRNAs and lncRNAs-targeted mRNAs to further understand the role of lncRNAs in the DDR. Interestingly, we found that the most enriched GO terms were significantly associated with the histone mRNA metabolic process, nucleosome organization, basic amino acid transport, nucleoplasm, and positive regulation of protein ubiquitination involved in ubiquitin-dependent protein catabolic process categories (Figure 4 and Figure 5). These results are partially consistent with previous studies. For example, Nie et al. found enrichment of genes involved in metabolism, the cell cycle and the DDR after X-ray irradiation [34]. Terradas et al. reported that after high-dose irradiation, lncRNAs involved in the cell cycle, DNA repair, and the DDR were identified [35]. However, in this study, we found that most enriched genes were also involved in histone regulation (Figure 5). This difference may be due to differences in the cell lines, radiation doses, and types of radiation used. In our study, 293T cells were treated with a high dose (8 Gy) of γ-radiation, which is highly toxic, and we harvested cells at 24 h after IR exposure. At this time, large-scale DNA damage may trigger histone changes in preparation for cells to enter the process of apoptosis. Similarly, previous studies have demonstrated that DNA damage leads to loss of large amounts of histone chromatin, which facilitates chromatin dynamics and recombination rates [31]. Wan et al. found that lncRNA JADE can upregulate the transcription of Jade1 which encodes a plant homeodomain (PHD) zinc finger protein. Upon DNA damage upregulation of Jade1 enhances histone H4 acetylation [36], Qian et al. observed acetylation-mediated proteasomal degradation of core histones during DNA repair [37]. Furthermore, several findings have suggested that ubiquitin can regulate protein interactions at DSB sites to promote accurate lesion repair [38]. Our research seems to be consistent with the above findings regarding biological functions. It is likely that some genes may regulate the DNA damage response through regulation of the histone mRNA metabolic process and regulation of protein ubiquitination.

According to the KEGG pathway analysis results, differentially expressed mRNAs and lncRNAs that target coding RNAs were mainly involved in Wnt signaling pathways and transcriptional dysregulation in cancer pathways. The Wnt signaling pathway is well-studied and engages in crosstalk with the DNA damage response [39]. Maintenance of genomic integrity after DNA damage mostly depends on activation of the tumor suppressor p53, which coordinates the roles of DNA repair systems and/or cell cycle checkpoints [40]. The Wnt signaling pathway is one of the main targets of p53 [41]. A recent study has demonstrated that the protein levels of Caudal type homeobox 2 (CDX2) is positively correlated with some DNA repair proteins. Meanwhile, CDX2 is also involved in liver metastasis of colorectal cancer as an important member of the Wnt signaling pathway [42]. In addition, genomic instability is a common feature of cancer cells and promotes the accumulation of oncogenic mutations [43]. Our findings indicated that cancer pathways are associated with the DNA damage response.

Recent studies have demonstrated that lncRNAs can act as microRNA (miRNA) sponges to regulate gene expression at the posttranscriptional level [44]. For example, lncRNA DANCR may act as a competing endogenous RNA to regulate Ras-related protein RAB1A expression by sponging miR-634 in glioma [45]. lncRNA HNF1A-AS1 acts as a competing endogenous RNA that promotes the progression of colon cancer metastasis by inhibiting the mir-34a/SIRT1/p53 feedback loop [46]. lncRNA HOST2 acts as an miRNA sponge to maintain the expression of oncogenes by isolating miRNA-let-7b, thereby maintaining the biological function of epithelial ovarian cancer [47]. Thus, we built a lncRNA–miRNA–mRNA network to further elucidate the possible functions of lncRNAs in the DDR. As described in Figure 7, lncRNA NR-038253 associates with 32 different miRNAs, and one miRNA can inhibit multiple mRNAs. Interestingly, we also found that some of the targeted mRNAs were involved in the radiation-induced DNA damage response. Shahrourpoly et al. found that poly(ADP-ribose) polymerase family member 10 (PARP10) played an important role in DNA damage repair [30]. HECW2 is an E3 ubiquitin protein ligase that regulates the stability of the AMOTL1 protein through lysine 63-linked polyubiquitination [48]. In addition, K63-linked polyubiquitination often occurs on the scaffolds histone H2A and proliferating cell nuclear antigen (PCNA) and recruits proteins that promote DNA repair [49]. Similarly, we found histone cluster 2 H2B family member E (HIST2H2BE) and H1 histone family member 0 (H1F0) associated with multiple miRNAs and mRNAs. This finding may suggest that histone loss is associated with the regulation of these lncRNAs. Taken together, our findings may deepen our knowledge of the molecular mechanism of the radiation-induced DDR.

## 4. Materials and Methods

### 4.1. Cell Culture

Human embryonic kidney cell line 293T (ATCC CRL-3216) were purchased from the American Type Culture Collection (Manassas, VA, USA) and cultured in Dulbecco′s modified eagle′s medium (Hyclone, Logan, UT, USA) supplemented with 10% fetal bovine serum (Gibco, Carlsbad, CA, USA), 2 mM l-glutamine and 1% penicillin/streptomycin at 37 °C in a humidified chamber with 5% CO_2_.

### 4.2. Irradiation of Cells

293T cells were divided into two groups: control group and irradiated group. A Gammacell-40 ^137^Cesium γ-ray irradiator (Atomic Energy of Canada, Chalk River, ON, Canada) was used to treat the irradiated group cells for a total dose of 8 Gy at 1 Gy/min at room temperature. The control group cells underwent the same condition as the irradiated group except for exposure to IR. Each group had three independent samples.

### 4.3. RNA Extraction

RNA was extracted from 293T cells of the control group and irradiation group at 24 h post-radiation using Trizol reagent (Invitrogen, Carlsbad, CA, USA) according to the manufacturer’s guidelines. The concentrations of total RNA were determined using a NanoDrop ND-2000 spectrophotometer (Thermo, Waltham, MA, USA). RNA quality was evaluated by using electrophoresis under denaturing conditions.

### 4.4. High-Throughput Sequencing

According to high-throughput sequencing results, transcriptome sequencing and subsequent bioinformatics analysis were carried out by Cloud-Seq Biotech (Shanghai, China). First the Ribo-Zero rRNA removal kits (Illumina, San Diego, CA, USA) was used to eliminate ribosomal RNA from the total RNA following the manufacturer’s instructions. Subsequently, RNA libraries were prepared by using rRNA-depleted RNA with the TruSeq Stranded Total RNA Library Prep Kit (Illumina) Sequencing libraries were assessed through quantification and quality control using a Bio Analyzer 2100 system (Agilent Technologies, Santa Clara, CA, USA). Then, 10 pM libraries were denatured into single-stranded DNA molecules, captured on Illumina flowcells, amplified in situ as clusters and finally sequenced for 150 cycles on an Illumina HiSeq 4000 Sequencer (Illumina).

### 4.5. Sequencing Analysis of lncRNA

According to the initial judgment and identification, paired-end reads were obtained from the Illumina HiSeq 4000 sequencer. Q30 was used for quality control. Additionally, cutadapt software was used for 3′ adaptor trimming and to remove low-quality reads [50]. The high-quality trimmed reads were used to analyze lncRNAs and mRNAs by mapping to the human reference genome (UCSC hg19) with HISAT2 software (v2.0.4 , Open Source Initiative (OSI), Palo Alto, CA, USA) [51].

### 4.6. Identification and Expression Analysis of lncRNAs and mRNAs

Cuffdiff software was used to obtain the fragments per kilobase of exon per million reads mapped (FPKM) values for the expression profiles of lncRNAs and mRNAs, and fold changes and *p*-values were calculated based on the FPKM values. We considered a fold change ≥ 2.0 with *p* < 0.05 and an FPKM value ≥ 0.1 in at least one sample from a group to indicate differentially expressed lncRNAs and mRNAs.

### 4.7. Quantitative Real-Time PCR

Quantitative real-time polymerase chain reaction (qRT-PCR) was performed to validate the expression of the lncRNAs. cDNAs were synthesized from total RNAs from the samples using the PrimeScript RT reagent kit (Perfect Real Time; TaKaRa, Osaka, Japan). For qPCR, EVA Green qPCR Mix was used (Abm, Vancouver, BC, Canada) with the BIO-RAD CFX Connect Real-Time PCR System (Bio-Rad, Hercules, CA, USA). Primers were designed based on the cDNA sequences and are listed in Table 1. *GAPDH* was employed as a reference for normalization. The cycling program consisted of enzyme activation for 10 min at 95 °C, heating denaturation for 15 s at 95 °C, annealing and extension for 60 s at 60 °C, with 40 cycles of denaturation, annealing and extension. The tests were performed with three independent samples, and each assay was conducted in triplicate. The relative expression of lncRNAs was represented by the 2^−ΔΔCt^ method.

### 4.8. GO Annotations and KEGG Enrichment

GO and KEGG analyses were applied to predict the biological functions of the lncRNA target genes and mRNAs. GO analysis was applied for the identification and annotation of the differentially expressed genes in the three categories of molecular functions, biological processes, and cellular component. KEGG enrichment analysis was used to analyze the important pathways involving the differentially expressed target genes and mRNAs of lncRNAs. The false discovery rate (FDR) was used to correct the *p*-values. A *p*-value < 0.05 was considered to indicate statistically significant enrichment.

### 4.9. lncRNA–miRNA–mRNA Network

To further understand the potential interactions of differentially expressed lncRNAs and mRNAs, we constructed a lncRNA–miRNA–mRNA network. According to popular miRNA target gene prediction software, the interactions between the different lncRNAs and miRNAs were predicted. Subsequently, miRNA binding site and target mRNA prediction was performed with proprietary software based on TargetScan (v7.0, Whitehead Institute, Cambridge, MA, USA) and miRanda (v3.3a, Memorial Sloan Kettering Cancer Center, New York, NY, USA). Thus, the lncRNA–miRNA–mRNA network was constructed using Cytoscape software (v3.1.0, National Institute of General Medical Sciences, Bethesda, MD, USA) on the basis of lncRNAs, miRNAs, and mRNAs.

## Figures and Tables

**Figure 1 ijms-20-02968-f001:**
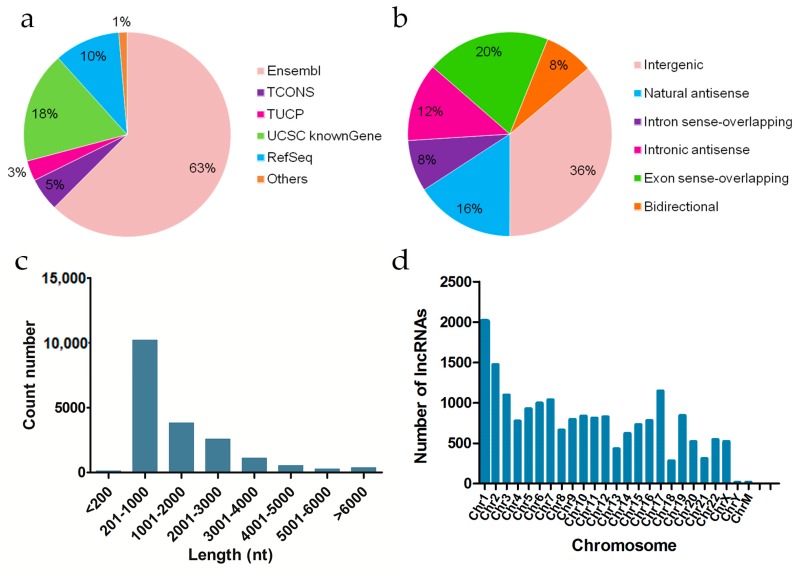
Expression signatures of long noncoding RNAs (lncRNAs) in 293T cells. (**a**) Pie chart showing the comparative numbers of lncRNAs from authoritative databases. (**b**) Pie chart showing the components of lncRNAs in each category according to their relative chromosomal position to coding genes. (**c**) Length distribution of lncRNAs. (**d**) Chromosome distribution of lncRNAs showing the numbers of lncRNAs located on different chromosomes.

**Figure 2 ijms-20-02968-f002:**
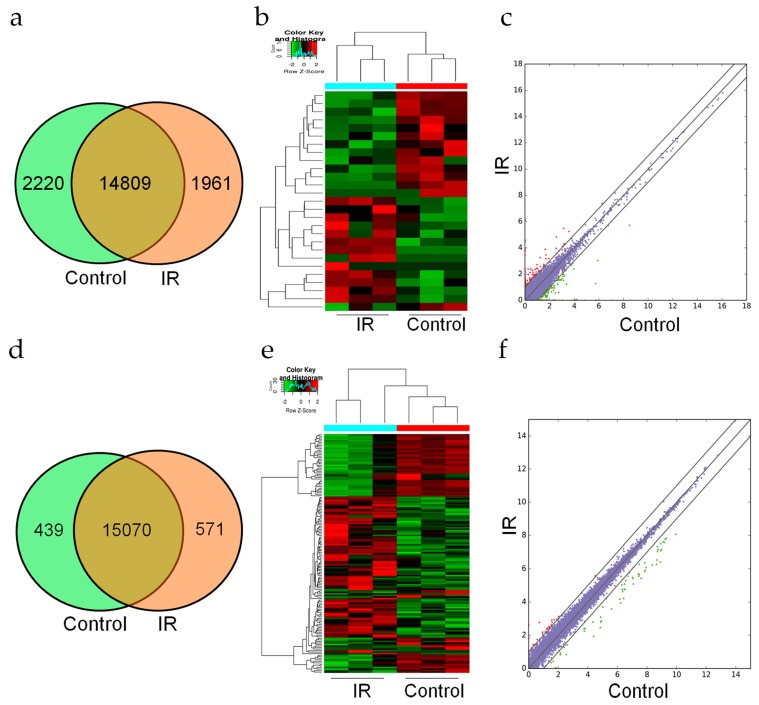
Expression profile changes in lncRNAs and mRNAs after exposure to radiation. (**a–c**) Venn diagram, heat map, and scatter plot of the changes in lncRNAs. (**d–f**) Venn diagram, heat map, and scatter plot of the changes in mRNA. The Venn diagrams show the number of overlapping lncRNAs or mRNAs in the irradiated and control groups. The heat maps show the hierarchical clustering of altered lncRNAs or mRNAs in the two groups. Red represents upregulation, and green represent downregulation. Scatter plots show the expression variation of lncRNAs or mRNAs between the two groups. Each lncRNA or mRNA is represented by a dot, with red dots indicating upregulation and green dots indicating downregulation. The purple dots in the center area indicate no difference (*p* < 0.05, fold change ≥ 2).

**Figure 3 ijms-20-02968-f003:**
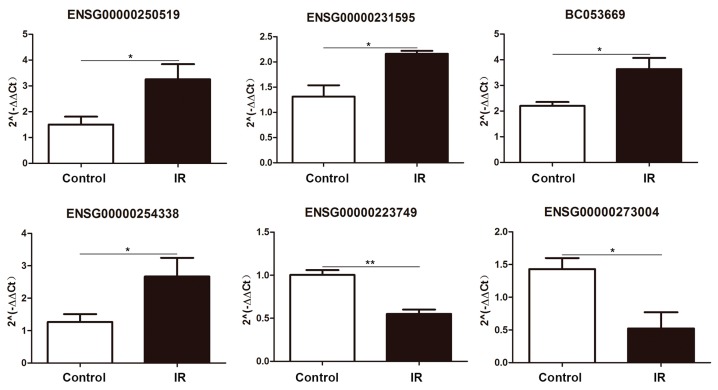
Validation of lncRNA expression. qRT-PCR validation of four upregulated and two downregulated lncRNAs. ENSG00000250519, ENSG00000231595, BC053669, and ENSG00000254338 were upregulated, and ENSG00000223749 and ENSG00000273004 were downregulated. * *p* < 0.05 and ** *p* < 0.01

**Figure 4 ijms-20-02968-f004:**
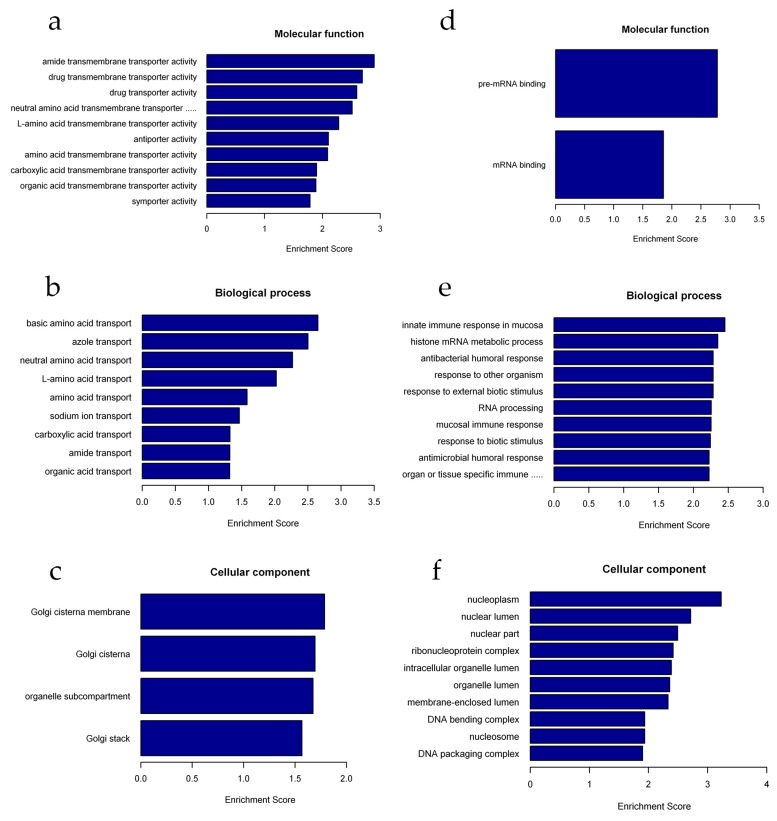
Gene ontology (GO) analysis of differentially expressed lncRNAs. (**a–c**) Molecular function (MF), biological process (BP), and cellular component (CC) terms of the upregulated lncRNA-adjacent coding genes. (**d–f**) Molecular function (MF), biological process (BP), and cellular component (CC) terms of the downregulated lncRNA-adjacent coding genes.

**Figure 5 ijms-20-02968-f005:**
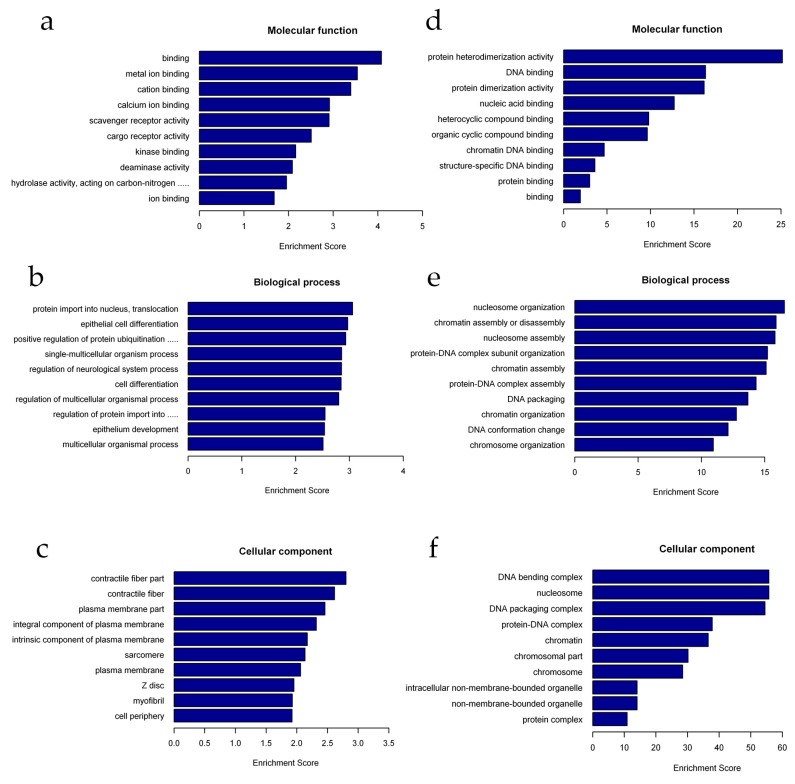
Gene ontology (GO) analysis of differentially expressed mRNAs. (**a–c**) Molecular function (MF), biological process (BP), and cellular component (CC) terms of the upregulated mRNAs. (**d–f**) Molecular function (MF), biological process (BP), and cellular component (CC) terms of the downregulated mRNAs.

**Figure 6 ijms-20-02968-f006:**
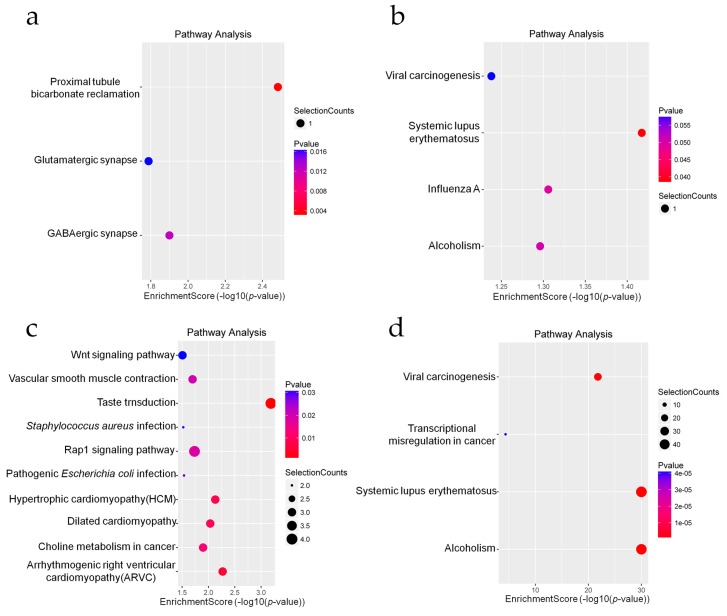
Kyoto Encyclopedia of Genes and Genomes (KEGG) signaling pathway analysis of differentially expressed genes. (**a**) Enriched KEGG pathways of upregulated lncRNA-adjacent coding genes. (**b**) Enriched KEGG pathways of downregulated lncRNA-adjacent coding genes. (**c**) Enriched KEGG pathways of upregulated mRNAs. (**d**) Enriched KEGG pathways of downregulated mRNAs.

**Figure 7 ijms-20-02968-f007:**
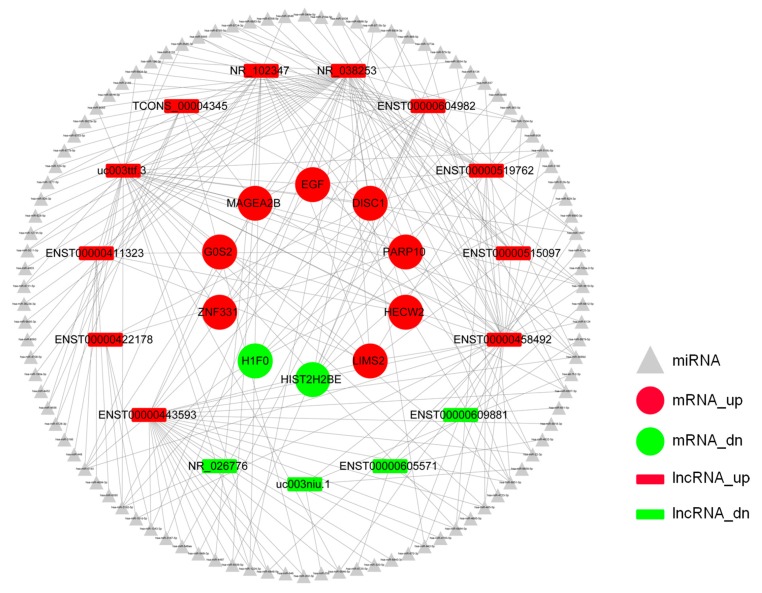
lncRNA–miRNA–mRNA network analysis of lncRNAs and targeted mRNAs. The rectangles represent lncRNAs; arrowheads represent miRNAs; circles represent mRNAs; red represents upregulation; and green represents downregulation.

**Table 1 ijms-20-02968-t001:** Primers used for qRT-PCR and validation.

Gene Name	Primer Sequence (5′–3′)	Amplicon (bp)
*ENSG00000250519*	F: CAATCAGCGAGACTCCGTGG R: CTGTGCTTTCTGGGCTTCTTT	259
*ENSG00000231595*	F: GATTACAGGCACCTACAACAGAAG R: CTTTGGCAGTTTGCTTCACGA	219
*BC053669*	F: GGAATGACACTGCCCGAAC R: TCTTTCAACCTTTCCCTCCAC	87
*ENSG00000254338*	F: GGATGTGATGGTTGGAGTGGA R: CGCTCTGGATTGAGGCTCTT	89
*ENSG00000223749*	F: CAGACAAGAACTAAAGTGGAACCC R: AACGGAAATCAAAAGCAGCA	97
*ENSG00000273004*	F: GTGGAGAGTGTGAGGAAGAAGAGA R: GTGAAAAGTGGAAGTAAACTGGTGT	173
*GAPDH*	F: CTCTGATTTGGTCGTATTGGG R: TGGAAGATGGTGATGGGATT	168

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
