# Peer review of "Integrated Analysis of the Altered lncRNAs and mRNAs Expression in 293T Cells after Ionizing Radiation Exposure"

_ijms, 2019, doi:10.3390/ijms20122968_

Reviewer 1 Report

Authors demonstrated lncRNA alteration after ionizing radiation exposure in human embryonic kidney 293T cells.

Overall science is fine. But I noticed many English problems. I recommend authors to get professional English correction before re-submission. Plus all authors should check English.

8 Gy of gamma-rays is very lethal. I believe survival fraction should be approximately 10%. RNA is isolated at 24 hours after irradiation. I think apoptosis or cell death may be started by this time. Please discuss this potential issue in the discussion part. Please explain clearly why analysis at 24 hours was chosen by authors.

Please change title end from radiation to ionizing radiation exposure.

Figure 4, 5, 6 is very difficult to read. Please fix them.

Cammacell-40....Gammacell-40, etc...

Author Response

Dear reviewer:

    Thank you for the insightful comments on our manuscript, and we have revised the manuscript to accommodate the requested changes.

    Respond to your comments point by point in a PDF file.

Reviewer 2 Report

In this paper, the authors conduct a bioinformatics study to identify lncRNAs and mRNAs that are differentially expressed in 293T cells after exposure to 8Gy irradiation.  The expression of several lncRNAs and mRNAs was determined by high throughput sequencing and six of the interesting lncRNAs were validated using qPCR.  GO and KEGG analysis was performed to identify the biological processes impacted after exposure to irradiation.  A limited lncRNA-miRNA-mRNA network was constructed.

I have limited enthusiasm for this limited bioinformatics study as there are several such reports already in the literature that are more useful clinically in cancer and disease.  The usefulness of determining which lncRNAs show expression changes in 293T cells is not clear.  Moreover, the authors should mention which databases they have used (Figure 1a) for their analysis.  How did the authors determine that 8 Gy was the correct dose of irradiation that should be used? The authors obtained 18990 lncRNAs from their samples but validated only 6 lncRNAs by qPCR.  How did they determine which of these lncRNAs was important in the DNA damage response?  The paper is very limited in its scope and the rationale for such a study in 293T cells is not clear.  Therefore I cannot recommend publication.

Author Response

Dear reviewer:

We appreciate your suggestions and comments on our manuscript, which helped us to improve our manuscript. We have carefully considered your suggestions and revised our manuscript accordingly.

A point-by-point response to your comments is listed below.

 Point 1: I have limited enthusiasm for this limited bioinformatics study as there are several such reports already in the literature that are more useful clinically in cancer and disease. The usefulness of determining which lncRNAs show expression changes in 293T cells is not clear.  

 Response 1: We are very sorry that our manuscript was not attractive to you. We agree that bioinformatics analysis of lncRNAs in cells or tissues of cancer and other diseases has important clinical significance. However, the regulatory functions of lncRNAs remain unclear in most fields, such as radiation-induced cellular responses. The reason we chose 293T cells for this study is 293T cells are one of the most commonly used cell lines in the studies of DNA damage, as some examples shown in the following listed papers. Therefore, 293T cells may be considered a model cell line in the field of DNA damage to get the general idea of cellular responses to ionizing radiation (IR) or other genotoxic stresses. We believe understanding the expression changes of lncRNAs in 293T cells exposed to IR will be helpful to reveal the unknown functions of lncRNAs.

1. Morris JR, Boutell C, Keppler M, et al. The SUMO modification pathway is involved in the BRCA1 response to genotoxic stress. Nature. 2009; 462(7275):886-890.

2. Bass TE, Luzwick JW, Kavanaugh G, et al. ETAA1 acts at stalled replication forks to maintain genome integrity. Nat Cell Biol. 2016; 18(11):1185-1195.

3. Inano S, Sato K, Katsuki Y, et al. RFWD3-Mediated Ubiquitination Promotes Timely Removal of Both RPA and RAD51 from DNA Damage Sites to Facilitate Homologous Recombination. Mol Cell. 2017; 66(5):622-634.

4. Xia J, Chiu LY, Nehring RB, et al. Bacteria-to-Human Protein Networks Reveal Origins of Endogenous DNA Damage. Cell. 2019; 176(1-2): 127–143.

Point 2: The authors should mention which databases they have used (Figure 1a) for their analysis.

Response 2: Thanks for your suggestion. We analyzed our data using Ensembl, UCSC knownGenes, TCONS, and TUCP databases, as well as some data reported in the papers. We have added this information on line 91 in the revised manuscript.

Point 3: How did the authors determine that 8 Gy was the correct dose of irradiation that should be used?

Response 3: Clonogenic survival assay is a classical method to investigate the cellular response to IR. The most commonly used doses are 2, 4, 6, 8 Gy in this assay. Due to the high cost of high-throughput RNA sequencing and our limited funding, we could choose only one dose of radiation for current research. Finally we chose the highest dose, 8 Gy, for our study because we predicted higher dose might result in more changes in lncRNAs. In addition, 8 Gy irradiation was also selected for the following research.

1. Wang K, Zhu M, Ye P, et al. Ionizing radiation-induced microRNA expression changes in cultured RGC-5 cells. Mol Med Rep. 2015; 12(3):4173-4178.

2. Li L, Liu H, Du L, et al. MiR-449a Suppresses LDHA-Mediated Glycolysis to Enhance the Sensitivity of Non-Small Cell Lung Cancer Cells to Ionizing Radiation. Oncol Res. 2017; 26(4):547-556.

3. Wang F, Cheng J, Liu D, et al. P53-participated cellular and molecular responses to irradiation are cell differentiation-determined in murine intestinal epithelium. Arch Biochem Biophys. 2014; 542(10):21-27.

4. Kote-Jarai Z, Salmon A, Mengitsu T, et al. Increased level of chromosomal damage after irradiation of lymphocytes from BRCA1 mutation carriers. Br J Cancer. 2006; 94(2):308-310.

Point 4: The authors obtained 18990 lncRNAs from their samples but validated only 6 lncRNAs by qPCR. How did they determine which of these lncRNAs was important in the DNA damage response?

Response 4: Our primary aim is to investigate the differentially expressed lncRNAs after IR treatment. So we randomly selected 6 lncRNAs from 49 differentially expressed lncRNAs for qPCR to validate their expression alteration. It’s usually only selected a few lncRNAs for verification, as shown in the following listed papers.

1. Li Y, Zhang C, Qin L, et al. Characterization of Critical Functions of Long Non-Coding RNAs and mRNAs in Rhabdomyosarcoma Cells and Mouse Skeletal Muscle Infected by Enterovirus 71 Using RNA-Seq. Viruses; 2018; 10(10): 556.

2. Pingsen Z, Sudong L, Zhixiong Z, et al. Analysis of expression profiles of long noncoding RNAs and mRNAs in brains of mice infected by rabies virus by RNA sequencing. Sci Rep. 2018; 8(1):11858.

3. Jie M, Li Z, Ping Y, et al. Integrated analysis of long noncoding RNA expression profiles in lymph node metastasis of hepatocellular carcinoma. Gene. 2018; 676: 47-55.

As for identifying the important lncRNAs in DDR, we conducted bioinformatic analyses of Gene Ontology (GO), Kyoto Encyclopedia of Genes and Genomes (KEGG) pathways, and lncRNA-miRNA-mRNA networks to determine it.

Point 5: The paper is very limited in its scope and the rationale for such a study in 293T cells is not clear.

Response 5: We agree that our study has limited clinical implications. However, we think it’s also important to explore the mechanistic details of how cells respond to ionizing radiation, especially in 293T cells, one of the most commonly used cell lines. Many of the major discoveries in biology have been studied in general model cell lines. Our study aims to provide general idea of cellular response to IR.

Round  2

Reviewer 2 Report

I remain unconvinced that the study has findings that will be of broad interest to readers.  However IJMS is an open access journal, so publication is not strictly based on novelty of findings. Under these criteria, the paper would be suitable for publication, however, the authors should add their rationale for 293T cells and references that they have noted in their response (Points 1 and 5). 

Author Response

Response to Reviewer 2 Comments

Point 1: I remain unconvinced that the study has findings that will be of broad interest to readers. However IJMS is an open access journal, so publication is not strictly based on novelty of findings. Under these criteria, the paper would be suitable for publication, however, the authors should add their rationale for 293T cells and references that they have noted in their response (Points 1 and 5).

Response 1: Thanks for your suggestion. As requested, we have added one sentence to introduce 293T cells and one sentence to give our rationale for studying 293T cells in the introduction section (Line 53-56) and cited references (Ref 7-13) in the revised manuscript.